# The Identification of the Mitochondrial DNA Polymerase γ (Mip1) of the Entomopathogenic Fungus *Metarhizium brunneum*

**DOI:** 10.3390/microorganisms12061052

**Published:** 2024-05-23

**Authors:** Stylianos P. Varassas, Sotiris Amillis, Katherine M. Pappas, Vassili N. Kouvelis

**Affiliations:** 1Section of Genetics and Biotechnology, Department of Biology, National and Kapodistrian University of Athens, 15784 Athens, Greece; svarassas@biol.uoa.gr (S.P.V.); kmpappas@biol.uoa.gr (K.M.P.); 2Section of Botany, Department of Biology, National and Kapodistrian University of Athens, 15784 Athens, Greece; samillis@biol.uoa.gr

**Keywords:** mitochondrial DNA polymerase γ–Mip1, mitochondrial DNA replication, under-expression of Mip1, entomopathogenic fungi, *Metarhizium brunneum*

## Abstract

Replication of the mitochondrial (mt) genome in filamentous fungi is under-studied, and knowledge is based mainly on data from yeasts and higher eukaryotes. In this study, the mitochondrial DNA polymerase γ (Mip1) of the entomopathogenic fungus *Metarhizium brunneum* is characterized and analyzed with disruption experiments and its in silico interactions with key proteins implicated in mt gene transcription, i.e., mt RNA polymerase Rpo41 and mt transcription factor Mtf1. Disruption of *mip*1 gene and its partial expression influences cell growth, morphology, germination and stress tolerance. A putative in silico model of Mip1-Rpo41-Mtf1, which is known to be needed for the initiation of replication, was proposed and helped to identify potential amino acid residues of Mip1 that interact with the Rpo41-Mtf1 complex. Moreover, the reduced expression of *mip*1 indicates that Mip1 is not required for efficient transcription but only for replication. Functional differences between the *M. brunneum* Mip1 and its counterparts from *Saccharomyces cerevisiae* and higher eukaryotes are discussed.

## 1. Introduction

DNA polymerase γ, also designated as Mip1 in fungi and most specifically in yeasts [1,2], plays the most important role in mitochondrial (mt) replication in fungi and consists of a single subunit, which is homologous to the catalytic subunit PolγA (POLG1) in humans [3]. To initiate the molecular process of replication of mtDNA, the complex Rpo41-Mtf1 is also required [4,5,6], since Rpo41, the mt RNA polymerase, when combined with the Transcription Factor 1 Mtf1, has activity of mitochondrial primase [7]. Both Mip1 and Rpo41 have homology with the corresponding genes of the bacteriophages T5 and T3/T7, respectively [8,9,10], but especially in the cases of mitochondrial DNA polymerases, like the yeast Mip1 and human POLG1, N- and C-terminal extensions (of different sizes), which have not found in bacterial and phage polymerases and have been identified and studied [11,12].

While the majority of research on mt genome diversity and its functions has been focused on yeasts among the fungi [13,14,15], it is important to expand studies to other subphyla of Ascomycota, and specifically to Pezizomycotina, which include model species, like *Neurospora crassa* and *Aspergillus nidulans*; biotechnologically important species, like the entomopathogenic fungi used as Biological Control Agents (BCAs) for the protection of crops; pathogens to humans, animals and plants, like *Aspergillus fumigatus*, *Beauveria bassiana* and *Fusarium oxysporum*, respectively; and symbionts, like the ones in lichens and mycorrhiza. Therefore, there is a need to fully clarify the similarities of the mt gene expression mechanisms in these species that are not putatively anaerobes, as yeasts are. Entomopathogenic fungi like the hypocrealean species *Metarhizium brunneum* may act as model organisms for studying the DNA replication mechanism of mt genes for two reasons. Firstly, the mt genome is already known and has been comparatively analyses within the genus *Metarhizium* [16] in addition to its whole-genome analysis [17]. Secondly, species of the fungal genus *Metarhizium* have been used as BCAs, as they have proven to be benign alternatives for the protection of several different crops worldwide [18,19,20]. Gaining insights into the functional mechanisms of their mt genomes (e.g., mtDNA replication) may provide a starting point for future genetic modifications of these genomes, with a final aim of improving the efficacy of their entomopathogenic activity against pests like aphids and whiteflies which destroy crops like maize, potatoes and tomatoes [18]. In this work, the *mip*1 gene from *M. brunneum* was isolated and partially inactivated in order to investigate its impact on the phenotype of the fungus. Additionally, an in silico analysis was performed in order to find the putative amino acid residues of Mip1 which interact with the Rpo41-Mtf1 complex in order to act as primase during replication.

## 2. Materials and Methods

### 2.1. Media, Strains and Growth Conditions

Complete (CM) and Minimal media (MM) containing carbon and nitrogen sources, as well as salt and trace element solutions, were used in liquid or solid form complemented with 1.5% (*w*/*v*) agar. Potato dextrose agar (PDA, AppliChem, Darmstadt, Germany) was also used as complete medium. In all cases, the pH was set at 6.5. Unless otherwise stated, glucose was used as a carbon source at a concentration of 1% (*w*/*v*). Sodium nitrate (NaNO_3_) was used as a nitrogen source at a final concentration of 10 mM. The salt solution, consisting of KCl, MgSO_4_ and KH_2_PO_4_, and the trace element solution consisting of Na_2_B_4_O_7_, CuSO_4_, FeO_4_P, MnSO_4_, Na_2_MoO_4_ and ZnSO_4_, were used as described in similar experiments of *Aspergillus nidulans* [URL: https://www.aspergillus.org.uk/lab_protocols/care-and-feeding-of-aspergillus-fgsc/ (accessed on 19 May 2024)] in the Fungal Genetics Stock Center (FGSC). Radial growth on solid MM was assessed at 25 °C, pH 6.5 for a time period of 3–10 days, whereas the growth of conidia suspensions in liquid MM for downstream applications was assessed at 25 °C, pH 6.5, 150 rpm for a time period of 12–24 h. *M. brunneum* strain ARSEF 3297 was used in all experiments. The *Escherichia coli* DH5α strain was used for cloning purposes.

### 2.2. Nucleic Acid Manipulations

Genomic DNA from *M. brunneum* was isolated according to the protocol described in Apostolaki et al. [21]. Plasmid purification and DNA gel extraction were performed using the Nucleospin Plasmid and the Nucleospin Extract II kits (Macherey-Nagel, Düren, Germany). Restriction enzymes, T4-ligases and phosphatases were from Takara Bio (Kusatsu, Japan). DNA sequences were determined by CeMIA SA (Larissa, Greece). Routine and high-fidelity PCR amplifications were performed using KAPA Taq DNA and Kapa HiFi polymerases (Kapa Biosystems, Merck SA, Athens, Greece). Oligonucleotides 5UTR-MIP1[ApaI]-F, 5UTR-MIP1[SpeI]-R, MTS-MIP1[SpeI]-[5]F and MTS-MIP1[NotI]-[3]R carrying specific enzyme adaptor sequences are presented in Table 1 and were used to amplify the relevant fragments from genomic DNA. The gene for hygromycin B phosphotransferase from *Klebsiella pneumoniae* was amplified from plasmid pAG32 [22] using oligonucleotides HYGrB-pAG32-[XbaI]-F and HYGrB-pAG32-[SpeI]-R (Table 1). These fragments were subsequently joined and cloned into the pGEM-T vector (Promega, Madison, WI, USA) using single-step ligation reactions. For the construction of a cassette overexpressing mip1, the relevant sequence of the *alc*Ap promoter of the putative alcohol dehydrogenase homologue in *M. brunneum* was amplified using oligonucleotides alc-prom-[SpeI]-F and alc-prom-[SpeI]-R (Table 1), and the resulting fragment was cloned in frame between *hyg*B and *mip*1 ORF. Targeted integrations of gene fusions were achieved by the amplification of those linear cassettes, also carrying the marker *hyg*B, resulting in resistance to hygromycin B.

### 2.3. Transformation

Transformation in *M. brunneum* was performed as described in Koukaki et al. [23] for *Aspergillus nidulans* with some modifications. In brief, a conidial suspension of approximately 108–109 conidia of *M. brunneum* isolated from five plates grown for 7 days on MM 25 °C, after being passed through a nylon filter (mesh size 75 μm) to separate conidia from other irrelevant mycelial structures, were cultured in MM at 25 °C/130 rpm for 12–14 h, until conidial germ tube emergence. Germlings were subsequently incubated for 5 h in 1.2 M MgSO_4_, 10 mM orthophosphate, pH 5.8 supplemented with 500 mg cell wall lysing enzymes (Sigma-Aldrich, St. Louis, MO, USA), followed by a brief incubation of the resulting protoplasts with 1–2 μg linear DNA cassette, in 1 M Sorbitol, 10 mM Tris-HCl pH 7.5 and 10 mM CaCl_2_ in the presence of 60% (*w*/*v*) PEG6000. Transformants were selected after incubation as an overlay on MM plates supplemented with 1 M Saccharose and 0.25–1 mg/mL Hygromycin B for 7 days at 25 °C. Transformants were verified by PCR analysis and sequencing.

### 2.4. Microscopy

Samples were incubated in 3 cm petri dishes on coverslips in liquid MM supplemented with glucose as carbon and NaNO_3_ as nitrogen sources at 23–30 °C for 0–24 h. Samples were observed on an Axioplan Zeiss microscope (Zeiss, Jena, Germany) with Nomarski interference contrast and the resulting images were acquired with a Zeiss-MRC5 digital camera, using the ZEN 2.0 software. Conidial length-width ratio was calculated using the ZEN 2.0 (graphics tab-adding annotations) and SigmaPlot 14.5 software [24].

### 2.5. Protein Molecular Modeling

Prediction of the Mip1 secondary structure was performed using the PSIPRED 4.0 Workbench (UCL-CS Bioinformatics, London, UK) [25]. For homology modeling of the Mip1 protein, the Hidden Markov Model-based tool HHPred [26] and MODELLER 9.25 [27] were used, based on the highly similar crystal structure of a bacteriophage T7 DNA replication complex (Protein Data Bank, PDB 1T7P) with 97.74% similarity and 1.8 × 10^−22^ E-value, as described previously [28]. EDock software (replica exchange MC simulation) was used to describe the folding of the above predicted Mip1 of *M. brunneum* ARSEF 3297 with a 10 base-pair DNA molecule (ligand) [29]. Using MergeStructs Plugin, Version 1.1 (URL: https://www.ks.uiuc.edu/Research/vmd/plugins/mergestructs/ accessed on 19 May 2024), a cryo-EM structure of yeast mitochondrial RNA polymerase transcription initiation complex (6YMW) [30] was combined with our predicted Mip1 structure. Protein structures were visualized and compared using PyMOL 2.4 (https://pymol.org). In all programs, the default parameters were used. 

### 2.6. RNA Extraction, First-Strand cDNA Synthesis and Quantitative Real-Time PCR

Total cellular RNA was isolated from 50–100 mg grinded mycelium using the TRIzol™ Reagent (Invitrogen, Carlsbad, CA, USA), following the manufacturer’s instructions. The isolated total RNA (1 μg verified for its purity and quantity using Nanodrop, Thermo Fischer Scientific, Waltham, MA, USA) was reverse transcribed in a 20 μL reaction mixture to generate single-stranded complementary DNA (cDNA) using 100 U SuperScript™ II Reverse Transcriptase (Invitrogen, Carlsbad, CA, USA), 40U RNaseOUT Recombinant Ribonuclease Inhibitor (Invitrogen, Carlsbad, CA, USA) and 10 μM gene-specific primer (Table 1). Reverse transcription was performed at 42 °C for 50 min, followed by enzyme inactivation at 70 °C for 15 min. As a control for cDNA synthesis efficiency, the cDNAs from different target RNAs and a negative control (DNAse-treated RNA, no reverse transcription) were used as a template for amplification with specific primers by PCR (KAPA HiFi HotStart ReadyMix PCR Kit, Kapa Biosystems, Merck SA, Athens, Greece). In addition, DNA fragments generated with KAPA HiFi HotStart DNA Polymerase (β-actin: 194 bp, DNA polymerase γ: 174 bp, ATP synthase F0 subunit 9: 220 bp) were used directly for blunt-end cloning in vector pBluescript II SK (Stratagene, Agilent Technologies, Santa Clara, CA, USA) and sequencing in both directions using the M13 universal primers. Amplicon sequences were analysed using the “Sequence Scanner v2” software (Thermo Fisher Scientific Inc., Waltham, MA, USA). qPCR assays were performed on the StepOne™ (Applied Biosystems, Carlsbad, CA, USA) using SYBR Green I dye for the quantification of the genes of interest. A 10 μL reaction mixture, containing 5 μL Kapa SYBR Fast Universal 2× qPCR master mix (Kapa Biosystems, Roche Diagnostics, Wilmington, USA), 10 ng of cDNA template and 200 nM of each specific primer, was used in a PCR protocol (95 °C for 3 min, followed by 40 cycles of denaturation at 95 °C for 15 s, primer annealing and extension at 60 °C for 1 min), followed by typical melt curve analyses to distinguish specific amplicons from non-specific products and/or primer dimers. All qPCR reactions were performed using two technical replications for each tested sample and target, and the average Ct of each duplicate was used in quantification analyses. Transcripts were detected for the mitochondrial ATP synthase subunit 9 (*atp*9) and the mitochondrial DNA polymerase γ (*mip*1). The housekeeping gene β-actin was used as a reference control for normalization. In detail, the Ct values obtained from our WT and mutant RNA samples were directly normalized to the housekeeping gene of β-actin and then compared. We assumed that the amplification efficiencies of the *mip*1 and the *atp*9 and β-actin genes were close to 100 percent (as we had a calibration curve slope of −3.32). Firstly, the difference between the Ct values (ΔCt) of the *mip*1, the *atp*9 and the β-actin genes were calculated for each experimental sample. Then, the difference in the ΔCt values between the experimental and control samples ΔΔCt was calculated. The fold-change in expression of the *mip*1 and *atp*9 between the two samples was then equal to 2^−ΔΔCt^. 

### 2.7. Statistical Analysis

Statistical significance of differences between compared datasets was assessed using the analysis of variance (ANOVA) test, following the evaluation of homogeneity of variance across samples (F-test *p* ≤ 0.05). Datasets were then subjected to means separation using Tukey’s honest significant difference (HSD) test. Scale bars were added using SigmaPlot14.5 software.

## 3. Results

### 3.1. Identification of M. brunneum Mip1

Using BlastP and the corresponding sequence of the DNA polymerase gamma of *S. cerevisiae* (NP_014975.2), Mip1 from *M. brunneum* named as MBR_07687 was identified in NCBI Genome with Accession Number NW_014574705.1 (48% amino acid identity, 92% similarity). The gene was amplified from the genomic DNA of the wild-type strain ARSEF 3297, cloned and sequenced. The size of the gene was found to be 3499 bp, including an intron of 154 bp, thus encoding a protein of 1144 amino acids (Figure 1A, Appendix A). The physical and chemical properties of *M. brunneum* Mip1 were analyzed in silico, showing a protein mass of 129.68 kDa and an isoelectric point of 8.158 (Figure 1B). The MitoProt II (v1.101) program revealed a signal sequence indicating that the identified Mip1 is indeed a mitochondrial matrix protein. Phylogenetic analysis of Mip1 of *M. brunneum* within Ascomycota (Appendix A) was also performed to establish evolutionary relationships and to identify conserved regions (Appendix A). The analysis revealed that the Mip1 gene shows similar phylogeny to that based on the ITS1-5.8S-ITS2 region (Appendix A), as species of the same order can be seen to cluster together. This result offered another indication that the *M. brunneum* Mip1 was correctly identified, since the Mip1 gene is essential for survival and hence conserved throughout time, following the evolution of the organisms carrying it.

### 3.2. Mip1 Has an Important Role in Fungal Cell Growth, Morphogenesis, Conidiation and Stress Tolerance

In order to study the function of Mip1 and the cellular differentiation caused by mtDNA–related mitochondrial dysfunction, we attempted to generate knock-out mutants, but no mutant was acquired, as complete inactivation of *mip*1 is most probably lethal for exclusively aerobic species like *M. brunneum*. Therefore, the knock-down mutants were constructed based on homologous recombination via the transformation of linear DNA cassettes that included, as a selection marker, the gene responsible for hygromycin B resistance (Figure 2A). 

The ethanol-inducible, glucose-repressible promoter alcAp, often used for controllable protein overexpression in fungi [31,32], was selected as the candidate regulatable promoter. However, after multiple attempts and analysis of dozens of transformants, we failed to obtain a mutant carrying a stable alcAp integration at the *mip*1 genomic locus, a result suggesting that wild-type expression levels of *mip*1 are crucial for survival, and/or that the efficiency of homologous recombination in the wild-type strain is extremely low. Nevertheless, using a similar construct that carried all DNA fragments except the alcAp sequence, we isolated a strain with pronounced altered colony morphology, which proved to carry a functional insertion of *hyg*B between the *mip*1 ORF and its native promoter region (Figure 2A) that would plausibly alter the expression of *mip*1. This insertion was verified with PCR amplification (see Appendix A). Indeed, this mutant, from onwards designated as Mip1^−^, showed a 60% reduction of expression compared to the wild-type ARSEF 3297, as shown by quantitative qPCR analysis (Figure 2B,C). Interestingly, the gene expression levels of the mitochondrial ATP synthase subunit 9 (*atp*9) appear less affected (Figure 2C). Both these genes were normalized against the housekeeping gene of β-actin (see Materials and Methods). 

The Mip1^−^ mutant exhibited significantly slower growth (*p*-value: 8.9 × 10^−4^), less compact aerial mycelium, and abnormal colony periphery, both on MM and on CM (Figure 3A,B and Appendix A) at pH 6.5 and 25 °C. Colonies of Mip1^−^ grown on CM, PDA and MM media containing various carbon resources (Appendix A) were smaller compared to the wild type. Growth of all the strains of *M. brunneum* was sparse in nutrient poor media (MM) but denser in nutrient rich media (Appendix A). Furthermore, pigment deposition in the *mip*1-disrupted strains was delayed by 19 days. The mutant exhibited significant phenotypic alterations regarding its asexual reproductive capacity. Finally, conidiation was significantly lower than in the wild type (*p*-value: 9.4 × 10^−7^; Appendix A) on various carbon sources tested.

In order to further investigate Mip1 disruption phenotypes in the context of altered colony morphology and decreased conidial yield, conidia of the wild-type strain and the Mip1^−^ mutant were observed by microscopy after 4–6 of germination. Overall, Mip1^−^ cells showed rather abnormal cell morphology (Figure 3C). The conidia were elongated and swollen, compared to wild-type cells (rod-like conidia), at pH 6.5 and 25 °C. The wild-type strain produced larger conidia (~7 μm) when compared to the Mip1^−^ strain (~6 μm). On the other hand, the average width of conidia was 3.0 ± 0.5 μm in the Mip1^−^ strain and 2.5 ± 0.4 μm in wild-type cells, exhibiting a marginal difference (Figure 3D). To gain a better understanding of the phenotype in the two strains, the length/width ratio of the mutant conidia was determined. This was 2.14 for Mip1^−^ versus 2.38 for the WT conidia, (s_wt_ = 0.41, s_mut_ = 0.23 and CV_wt_ = 17% > CV_mut_ = 10%), resulting in smaller, more globose-elliptical conidia (Figure 3C,D).

The optimal growth temperature and pH of the WT strain was 25 °C and 6.5, respectively. It was found that the colony of the Mip1^−^ strain was smaller than that of the WT strain, the relative inhibition rate of growth of the Mip1^−^ strain increased significantly compared to the wild-type strain, and the hyphal growth of the Mip1^−^ strain was significantly decreased at various temperatures (Appendix A) in comparison to the WT strain. The conidial yield of the Mip1^−^ strain was much lower compared with the WT strain (Appendix A). Furthermore, the results showed that the germination rate of Mip1^−^ strain was slower compared to the WT strain (Appendix A). More specifically, differences in germination rate occurred mainly at 13 and 15 h, respectively (Appendix A). The GT50 (the mean 50% germination time) of the Mip1^−^ strain (15.94 ± 0.19 h) was moderately prolonged compared with the WT (13.45 ± 0.23 h) at 25 °C (Appendix A).

### 3.3. In Silico Analysis of M. brunneum Mip1 and Interactions with Rpo41-Mtf1 Complex

The in vitro re-constituted mtDNA transcriptional complexes (Rpo41–Mtf1) of *S. cerevisiae* and humans, aided with high-resolution structures and biochemical characterizations, have provided a deeper understanding of the mechanism and regulation of mitochondrial DNA transcription [30,33,34,35]. However, the study of the mechanism of the initiation of mtDNA replication at the corresponding level is limited [36]. A 3-D model of the Mip1-Rpo41-Mtf1 (see Materials and Methods) may be proposed for the initiation of mitochondrial DNA replication of *M. brunneum* based on our in silico analysis (Figure 1C, Appendix A). This proposition relies (i) on the fact that the mtDNA polymerase requires a primer and cannot initiate synthesis de novo [30,36], and (ii) on the hypothesis that Rpo41 synthesizes primers that are used for primer extension by Mip1 at origins of replication (Appendix A). 

The predicted Mtf1 interaction and binding with Rpo41 probably caused a change in the conformation of Rpo41 and Mtf1 to facilitate DNA melting, where the mtDNA (Appendix A, transcription bubble region shown in yellow) was melted from −4 to +2 [30]. Both Rpo41 and Mip1 of *M. brunneum* had four amino acids, which may have been responsible for a conserved conformation as our analysis from EV-couplings and ET-viewer indicates (Figure 1C, Appendix A) and, to an extent, they may play a role in the overall activity of the complex. 

## 4. Discussion

The fungal Mip1 is solely responsible for fast and faithful replication of the mitochondrial genome [2]. In *M. brunneum*, Mip1 is also identified as the DNA polymerase responsible for replication, and cells lacking a fully expressed Mip1 exhibit a slow-growing “petite-colony” phenotype. Mip1^−^ partially deficient cells not only display abnormal cell morphology but also show reduced growth rate. The Mip1^−^ phenotype in *M. brunneum* is similar to the phenotypes reported for *Schizosaccharomyces pombe* and reflects a relative tolerance for loss of mitochondria in both these two organisms, in contrast to *S. cerevisiae* which can afford losing all mitochondria [37,38]. The yeast *S. cerevisiae* is one of the few eukaryotic organisms that can either survive in the absence of mtDNA (rho0) or contain deletions in their mtDNA (rho^−^) [39]. Needless to say, Mip1 is important to support normal growth in *S. pombe* and *M. brunneum*, and cells bearing a *mip*1 inactivation exhibit different growth phenotypes and similar cell morphologies [6].

The qPCR experiment of this study showed that gene expression levels of the mitochondrial ATP synthase subunit 9 (*atp*9), which is co-transcribed with genes *nad*2, *nad*3 and the tRNAs lying upstream the gene *nad*2 in *M. brunneum*, i.e., the polycistronic transcription unit TU2 of the wild-type strain [10], are comparable to that of the Mip1^−^ mutant (Figure 2). While it seems expected that in a mutant for Mip1, both replication and transcription must be affected, since replication will not provide the same amount of mtDNA with that of the wild-type strain and in extent transcription of mt genes like *atp*9 will be reduced, it was found that the levels of expression of this mt gene remained significantly the same in both strains. Therefore, the hypothesis is that mt transcription is not affected from the partial inactivation of Mip1, but replication is undermined, as the reduction of the Mip1 expression and the slower growth of the mutant strains showed. This hypothesis can be indirectly supported further, by the mutants produced in the respective Mip1 of *S. cerevisiae* and more specifically in its C- terminal extension region of this yeast, as these mutants showed not a loss of function but a decrease in polymerization [11,12]. Moreover, it may be suggested that mt transcripts of the Mip1^−^ mutant of *M. brunneum* may have a longer half-life than those of the WT strain, or that there are other DNA polymerases, which may compensate the reduced function of Mip1. This latter hypothesis was based mostly on the presence of two DNA polymerases, i.e., Rev1p and Pol ζ, in *S. cerevisiae*, and the possible complementation of function with the existence of Rev1p, when Pol ζ is missed or inactive [40].

Direct interactions between Mip1 and Rpo41, the non-template strand at origins of replication and promoter sequences have been observed in *S. cerevisiae* [36,41,42,43]. A putative structural model of Mip1-Rpo41-Mtf1 complex was proposed in this work for initiation of mtDNA replication, since origins of replication are located at the same regions with promoters of transcription [36,44]. Mip1 may contact promoter DNA near the transcription start site at origin of replication through interaction with Rpo41-Mtf1 [45,46] and the priming nucleotide (Figure 1, Appendix A). This model structure is in agreement with the in vivo replication mechanism in *Candida albicans*, which involves a combination of the double-stranded break model for replicating the leading strand and Rpo41 as primase to replicate the lagging strand in this yeast [47].

## 5. Conclusions

This study underscores the pivotal role of the fungal Mip1 in mitochondrial genome replication besides the already known data from *S. cerevisiae* and other yeasts, as demonstrated through the slow growth phenotype of *M. brunneum* cells which have a partially impaired functional Mip1. This deficiency affects both the cell morphology and growth rate, akin to observations in other organisms like *S. pombe*. Interestingly, despite impaired replication in the Mip1-deficient strain, the qPCR analysis revealed comparable expression levels of mitochondrial genes, suggesting that while replication is compromised, transcription remains largely unaffected. The proposed structural model of the Mip1-Rpo41-Mtf1 complex may shed further light on the initiation of mtDNA replication, highlighting potential interactions crucial for this process. However, further research focusing on the structure and function of the mitochondria themselves is needed in the future, and these experiments into compensatory mechanisms and structural insights promise a deeper understanding of mitochondrial biology. This study represents the first significant step towards the usage of the hyphomycete *M. brunneum* as an alternative and complementary model system for molecular, genetic and biochemical studies of mitochondrial DNA replication in filamentous ascomycetes. Furthermore, this is the first approach to study the cellular function of Mip1 in *M. brunneum*, an entomopathogenic fungus used as BCA.

## Figures and Tables

**Figure 1 microorganisms-12-01052-f001:**
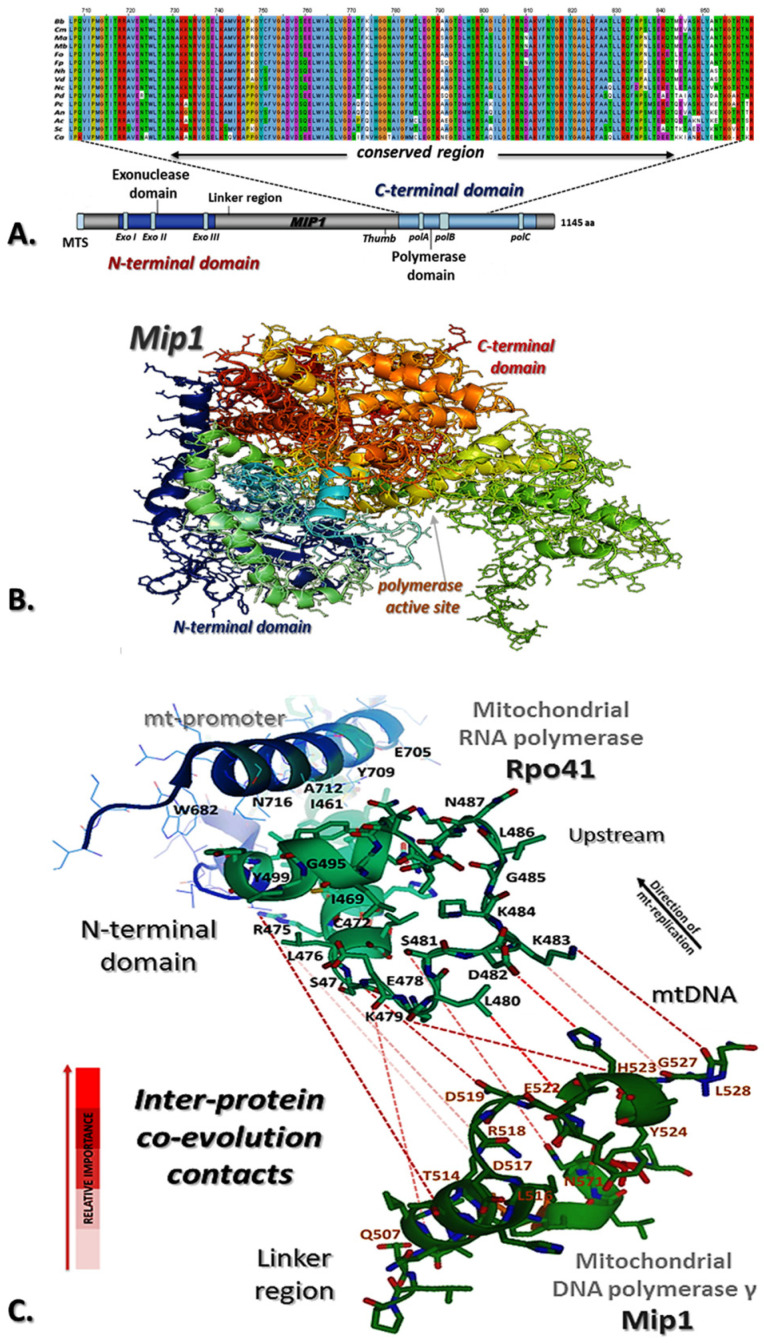
**Prediction of the structure of mitochondrial DNA polymerase in *M. brunneum* and inter-protein contacts for the complex of Rpo41-Mip1.** (**A**) The conserved polymerase activity domain as produced in amino acids alignments of the Mip1 (*M. brunneum* ARSEF 3297) with its orthologues in other fungal species and the location of this domain in the structure of the sequenced gene from ARSEF 3297. (**B**) Prediction of Mip1 structure in *M. brunneum* ARSEF 3297, based on crystallographic structure of the DNA polymerase of bacteriophage T7 (PDB_1T7P). (**C**) Prediction of evolutionary couplings between mt-RNA polymerase (Rpo41, N-terminal domain) and mt-DNA polymerase (Mip1, Linker region). The predicted inter-ECs for this complex were obtained based on a combination of the EV couplings server (Appendix A), the distance between interacting residues in protein complex (Rpo41-Mip1) from the combined structure of Mip1-Rpo41-Mtf1 complex for initiation of mtDNA replication using PyMol,and the characteristics-properties of these amino acids [monomer subunit in petrol-green (Rpo41, N-terminal domain) and deep-green (Mip1, Linker region), inter-ECs in red dashed lines based on the relative importance of ECs.

**Figure 2 microorganisms-12-01052-f002:**
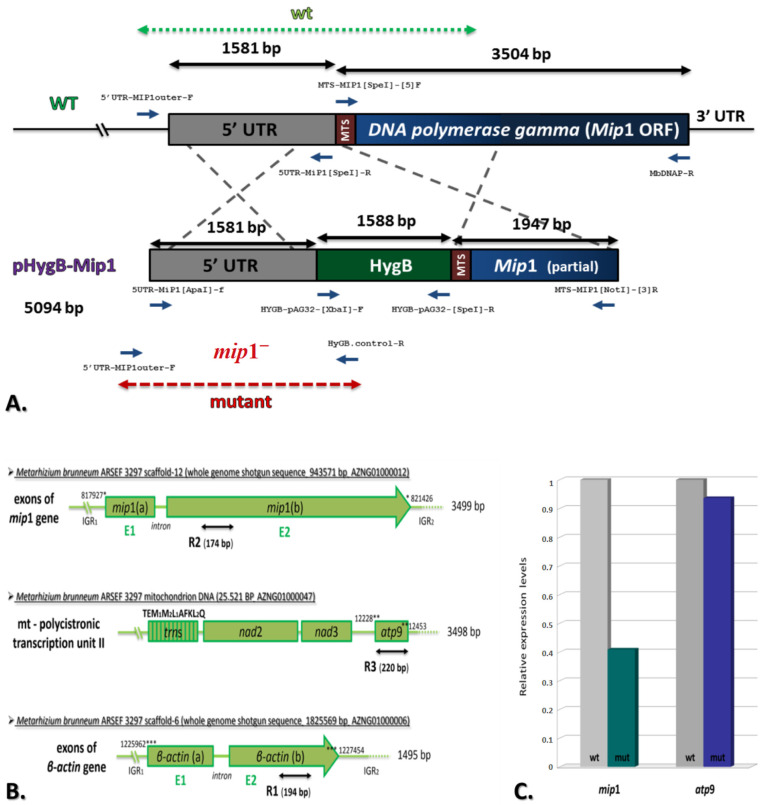
**Disruption of the *mip*1 gene in *M. brunneum* ARSEF 3297.** (**A**) Schematic representation of the Mip1 wild-type (WT) locus and the plasmid pHygB-Mip1 (containing two regions homologous to the mip1 reading frame) that were used for gene disruption through double crossover recombination. Replacement and WT-specific primer combinations and expected fragments are shown as grey lines. (**B**) Physical map of the *mip*1, *atp*9 and β-actin genes of *M. brunneum* strain ARSEF 3297 with locations of the primer sets R1, R2 and R3 (Table 1) for RT-PCR analysis of the above three genes in the WT and Mip1^−^ strains. Numbers with asterisks (*, **, ***) indicate their positioning in different scaffolds. (**C**) *mip*1 and *atp*9 expression were verified by Real-time RT-PCR using cDNA from wild-type *M. brunneum* ARSEF 3297 and mutant (Mip1^−^) strains (*p*-value ≤ 0.01, error bars ±1%-not shown). The β-actin gene was used to normalize the quantification of expression.

**Figure 3 microorganisms-12-01052-f003:**
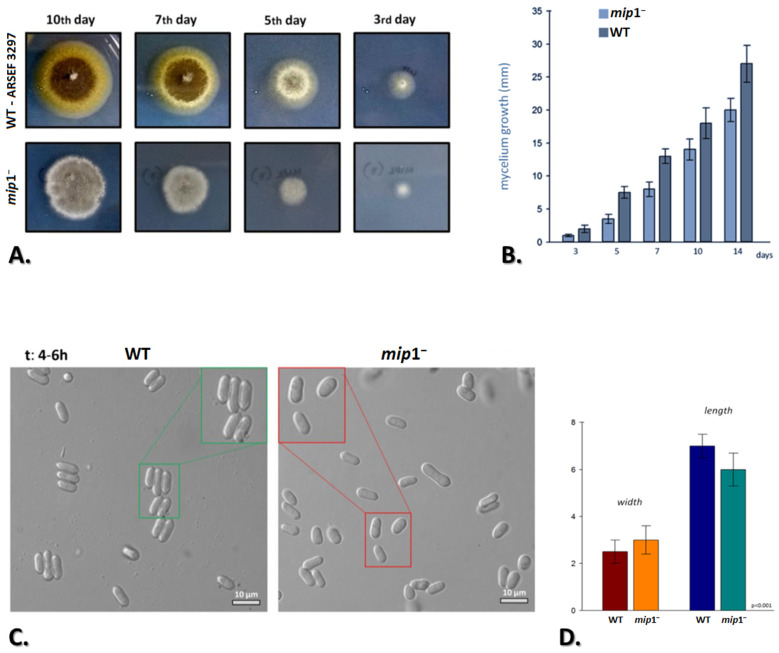
**Morphological characterization of Mip1^−^ strain as Mip1 regulates conidial differentiation.** (**A**) Colony morphologies of the M. brunneum wild-type isolate (ARSEF 3297) and the mutant Mip1^−^ after growth for 3, 5, 7 and 10 days on MM. (**B**) Colony diameter of the WT and Mip1^−^ strains on MM medium and cultured at 25 °C for 3–14 days. (**C**) Conidium differentiation in M. brunneum wild-type (normal/rod-like conidia) and Mip1^−^ mutant (swollen conidia) strains. (**D**) Average measurements (width and length in μm) from the two strains. All experiments were performed in triplicate. Bars: SD—statistical significance of differences was tested by one-way ANOVA, followed by Tukey’s post hoc test (*p* ≤ 0.001).

**Table 1 microorganisms-12-01052-t001:** Oligonucleotides used in this study for cloning and sequencing purposes. Underlined is the relevant restriction enzyme sequence.

Name	Sequence
5UTR-MIP1[ApaI]-F	CGCGGGGCCCTTCACATTGACATGTCATTAATGACTGCC
5UTR-MIP1[SpeI]-R	CGCGACTAGTCAAGAGTTGAACATCAAGTTTGAGCTGTC
HYGrB-pAG32-[XbaI]-F	CGCGTCTAGAGACATGGAGGCCCAGAATACCCTC
HYGrB-pAG32-[SpeI]-R	CGCGACTAGTCAGTATAGCGACCAGCATTCACATAC
alc-prom-[SpeI]-F	CGCGACTAGTCAATGAAGCCCATTCATCTTCTTGTCGACGAGC
alc-prom-[SpeI]-R	CGCGACTAGTTTGTCGGCTGTTTCGTGGCAAGTCGTG
MTS-MIP1[SpeI]-[5]F	CGCGACTAGTATGAATACGCTTTGTCCTGCTGCTGGTCACG
MTS-MIP1[NotI]-[3]R	CGCGGCGGCCGCCATACTCGTATTCCGACGACAGTGTGCC
β-actin-F	CATACATGGTCGAGAACAAGTCC
β-actin-R	AGTCCAGCGCCCCAAATAAC
mip1-F	AGGAGCGGGTGGTTGTTG
mip1-R	TCGCGATTCTTCTTGTGCC
atp9-F	ATGTTACAATCTTCAAAAATAATAGGAGC
atp9-R	TTAAGCAACATTTAATAATAATAATGACA

## Data Availability

The *M. brunneum* mutant strain Mip1^−^ is available upon request.

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
