# Peer review of "The Identification of the Mitochondrial DNA Polymerase γ (Mip1) of the Entomopathogenic Fungus Metarhizium brunneum"

_microorganisms, 2024, doi:10.3390/microorganisms12061052_

Round 1

Reviewer 1 Report

Comments and Suggestions for Authors

The statement in lines 224-225 are very confusing. Please rewrite to improve the clarity of the content.

The statement in lines 256-257 seems odd. The difference is really significant?

Comments on the Quality of English Language

Style and sentence construction need to be checked in a few places.

Author Response

Pleas see the attachement.

Reviewer 2 Report

Comments and Suggestions for Authors

The article is devoted to the identification of the gene encoding mitochondrial DNA polymerase (Mip1) in the entomopathogenic fungus Metarhizium brunneum, characterization of the phenotype of a strain with knockdown of this gene, and an in silico attempt to analyze the probable structure of the mtDNA replication complex in which Mip1 is possibly involved.

Major comments

1) To relate the molecular data to the phenotype of the M. brunneum mutant, some mitochondrial characteristics are needed. Do the authors have data on the rate of mtDNA replication in the mutant strain? Is the number and morphology of mitochondria altered in the mutant strain? How is mitochondrial function affected by Mip1 knockdown? Without these data, the authors cannot conclude a systematic study of Mip1 cellular functions.

2) Although the authors proposed a structural model of the Mip1-Rpo41-Mtf1 complex on the basis of high similarity, they did not provide any experimental evidence to support the correctness of the resulting model. Consequently, they cannot draw important and meaningful conclusions about the structure, function, and mechanism of the complex on the basis of bioinformatic analysis alone. The authors should either be extremely cautious in describing the probable mechanism of the complex or provide experimental data to support the correctness of the proposed structure as well as the mitochondrial DNA replication scheme.

Minor comments

Please indicate how you verified the correctness of the DNA fragments cloned into plasmids?

Line 125. In the phrase "with 97.74% probability" did you mean similarity or identity? Please clarify this.

In Section 2.5 "Molecular Modeling of Proteins", when describing the programs used, please indicate whether you customized some of the parameters or used the default ones.

In section 2.6, please specify how you checked the quality (integrity) of the purified RNA and its quantity.

It seems that primers for RT-PCR detection of atp9 mRNA are missing in Table 1 or that long primers atp9-F/atp9-R were used for RT-PCR? Please clarify this.

Please provide a mathematical method for quantifying mRNA levels by RT-PCR.

Figure 2C - Please add error bars and indicate which errors they reflect. Also indicate where statistically significant differences are observed.

Figure 3B and 3C. Please indicate where statistically significant differences are observed.

Comments on the Quality of English Language

English is good.

Round 2

Reviewer 2 Report

Comments and Suggestions for Authors

The authors have greatly improved the manuscript and have corrected the conclusions according to all the comments. I have no further significant comments and suggest that the manuscript be accepted for publication in the journal.